# IP_3_R-Mediated Calcium Release Promotes Ferroptotic Death in SH-SY5Y Neuroblastoma Cells

**DOI:** 10.3390/antiox13020196

**Published:** 2024-02-04

**Authors:** Joaquín Campos, Silvia Gleitze, Cecilia Hidalgo, Marco T. Núñez

**Affiliations:** 1Chica and Heinz Schaller Foundation, Institute for Anatomy and Cell Biology, University of Heidelberg, 69120 Heidelberg, Germany; joaquin.campostein@gmail.com; 2Biomedical Neuroscience Institute, Faculty of Medicine, Universidad de Chile, Santiago 8380000, Chile; silviagleitze@gmail.com (S.G.); chidalgo@uchile.cl (C.H.); 3Department of Neuroscience, Faculty of Medicine, Universidad de Chile, Santiago 8380000, Chile; 4Physiology and Biophysics Program, Institute of Biomedical Sciences and Center for Exercise, Metabolism and Cancer Studies, Faculty of Medicine, Universidad de Chile, Santiago 8380000, Chile; 5Department of Biology, Faculty of Sciences, Universidad de Chile, Santiago 7800024, Chile

**Keywords:** cell death, ferroptosis, calcium signaling, reactive oxygen species, endoplasmic reticulum, lipid peroxidation, oxidative stress, glutathione peroxidase, RSL3

## Abstract

Ferroptosis is an iron-dependent cell death pathway that involves the depletion of intracellular glutathione (GSH) levels and iron-mediated lipid peroxidation. Ferroptosis is experimentally caused by the inhibition of the cystine/glutamate antiporter xCT, which depletes cells of GSH, or by inhibition of glutathione peroxidase 4 (GPx4), a key regulator of lipid peroxidation. The events that occur between GPx4 inhibition and the execution of ferroptotic cell death are currently a matter of active research. Previous work has shown that calcium release from the endoplasmic reticulum (ER) mediated by ryanodine receptor (RyR) channels contributes to ferroptosis-induced cell death in primary hippocampal neurons. Here, we used SH-SY5Y neuroblastoma cells, which do not express RyR channels, to test if calcium release mediated by the inositol 1,4,5-trisphosphate receptor (IP_3_R) channel plays a role in this process. We show that treatment with RAS Selective Lethal Compound 3 (RSL3), a GPx4 inhibitor, enhanced reactive oxygen species (ROS) generation, increased cytoplasmic and mitochondrial calcium levels, increased lipid peroxidation, and caused cell death. The RSL3-induced calcium signals were inhibited by Xestospongin B, a specific inhibitor of the ER-resident IP_3_R calcium channel, by decreasing IP_3_R levels with carbachol and by IP_3_R1 knockdown, which also prevented the changes in cell morphology toward roundness induced by RSL3. Intracellular calcium chelation by incubation with BAPTA-AM inhibited RSL3-induced calcium signals, which were not affected by extracellular calcium depletion. We propose that GPx4 inhibition activates IP_3_R-mediated calcium release in SH-SY5Y cells, leading to increased cytoplasmic and mitochondrial calcium levels, which, in turn, stimulate ROS production and induce lipid peroxidation and cell death in a noxious positive feedback cycle.

## 1. Introduction

A major review, originating from the Nomenclature Committee on Cell Death, described 12 different types of cell death, which include intrinsic apoptosis, extrinsic apoptosis, membrane permeability transition (MPT)-dependent necrosis, necroptosis, ferroptosis, pyroptosis, parthanatos, entotic cell death, NETotic cell death, lysosome-dependent cell death, autophagy-dependent cell death, and immunogenic cell death [1]. Within these cell death processes, ferroptosis stands out, with over 4000 publications during 2023 alone.

Ferroptosis is a cell death process characterized by the accumulation of lipid peroxides in a manner that is dependent on cellular iron content [2,3]. Its importance, and the recent attention it has received, is related to the participation of ferroptosis in some of the most relevant of today pathologies, such as cancer and neurodegeneration [4,5]. Lately, the use of ferroptosis as a cell killing mechanism has brought about new strategies for the treatment of various malignant diseases [6].

The initial characterization of ferroptosis began in the context of a study using high-throughput screening to search for molecules with anti-cancer activity [7]; among other agents, erastin was found to produce non-apoptotic cell death much more efficiently in cells with the active RAS oncogene. Later insights into the effects of erastin determined its binding to mitochondrial voltage-gated anion channels (VDACs) [8] and excluded canonical cell death mechanisms, such as apoptosis, autophagic death, and necroptosis. Erastin-induced death is morphologically different from the classic types of cell death; it involves neither nuclear fragmentation nor vacuolization, and it entails a reduction in cytoplasm size, mitochondrial swelling, and plasma membrane rupture [8,9].

After these first studies, other compounds were found that induce the same type of cell death, with identical morphological characteristics and insensitivity to inhibitors, among others; the most notable among them was RSL3 [3]. Erastin and RSL3 induce a type of death with oxidative characteristics, which depends on iron, and it is characterized by the accumulation of lipid peroxides [9]. In particular, erastin, in addition to binding to VDAC channels, blocks the cystine/glutamate exchanger xCT, which leads to cellular GSH depletion [2,10]. This decrease in cellular GSH content leads to an oxidative imbalance that, among other factors, results in the inhibition of GPx4, which is the key enzyme that catalyzes the neutralization of membrane lipid peroxides to their respective alcohols. In contrast, RSL3 directly inhibits GPx4, resulting in the same accumulation of lipid peroxides but without decreasing GSH concentration [11,12,13,14].

Of note is the relationship between the inhibition of GPx4 and increased ROS production. A mediator between these two landmarks of ferroptosis is the pro-apoptotic protein Bid [15]; based on the observation that Bid inhibition preserves cell viability in RSL3-treated cells, the authors propose a sequence of events in which GPx4 inhibition enhances 12/15 LOX activity, which results in lipid peroxide formation and BID transactivation. In turn, activated Bid binds to mitochondria, decreasing their membrane potential and increasing ROS production [15]. Some research groups consider that ferroptosis is equivalent to oxytosis, a type of cell death described in hippocampal neurons [16], which was previously reported as a form of glutamate-mediated excitotoxicity [10]. In oxytosis, cell death may be initiated in a way similar to ferroptosis, that is, by blocking the xCT system. Since the xCT system is an exchanger-type co-transporter that uses the glutamate gradient to allow for cystine entry, when glutamate is present in excess (>5 mM) in the extracellular medium, it stops this gradient and inhibits cystine entrance. Since cystine is a GSH precursor, GSH concentrations decrease, which results in LOX activation, ROS generation, lipid peroxidation, and cell death. Death occurs in the absence of caspase activation, and there is protection by iron chelators. However, unlike what has been reported for ferroptosis, the execution of oxytosis depends on the massive influx of calcium from the extracellular medium [17]. The resulting intracellular calcium increase affects the mitochondria, producing the release of apoptosis-inducing factor (AIF), which can then translocate to the nucleus [18].

The role of calcium in ferroptosis was dismissed by pioneering studies, due to the lack of protective effects of CoCl_2_ and GdCl_3_, which are both plasma membrane calcium channels blockers [19]. A report on HT-1080 fibrosarcoma cells [2] also dismissed the role of calcium in ferroptosis, arguing that neither the calcium indicator fura-2 nor intracellular calcium chelation with BAPTA-AM showed protection. In contrast, Maher’s group proposed the hypothesis that ferroptosis and oxytosis are the same phenomenon, reporting the protective effect of CoCl_2_ against RSL3- and erastin-induced cell death, in a manner analogous to that classically shown for oxytosis [20,21]. In the same line, protective effects of the calcium chelator BAPTA and of CoCl_2_ against erastin-induced cell death have been reported in HT22 and LUHMES [22].

Ferroptosis is accompanied by increased ROS levels; interestingly, many components that regulate intracellular calcium levels are stimulated by ROS, including the ER-resident calcium channels IP_3_R and RyR [23,24,25,26]. Thus, it is conceivable that the resulting calcium increase, originating from ROS-stimulated calcium release from the ER, could contribute to ferroptosis. In this line, a recent work reported that the suppression of calcium release mediated by RyR channels, through the use of inhibitory concentrations of ryanodine, offers significant but partial protection against RSL3-induced cell death in hippocampal neurons [27]. However, the putative role of IP_3_R-mediated calcium release in ferroptosis is unknown. Given this background, the role of calcium in ferroptosis is clearly a matter of debate and may have nuances that depend on the cell system studied.

Here, we explored the hypothesis that in human neuroblastoma cells that do not express RyR channels, inhibition of the GPx4 enzyme activates IP_3_R-mediated calcium release, which results in increased cytoplasmic and mitochondrial calcium levels that contribute to the development of ferroptotic cell death.

## 2. Materials and Methods

### 2.1. Cell Culture

SH-SY5Y cells (ATCC, CRL-2266) were cultured at 37 °C and 5% CO_2_ in DMEM/F12 media (Gibco, New York, NY, USA) supplemented with 10% FBS and 1% PenStrep (Gibco). For experiments, 80% confluence cultures were used.

### 2.2. Cell Viability

Viability was assessed with the Vybrant^®^ MTT Cell Proliferation Assay Kit (V-13154; Thermo Fisher Scientific, Waltham, MA, USA) following the instructions of the manufacturer. Alternatively, cells were tested for propidium iodide (PI) permeability by incubation for 15 minutes (min) with 1 mg/mL of the fluorescence marker PI (Thermo Fisher Scientific) in modified Krebs buffer. Afterwards, PI red fluorescence was detected in random fields with a Nikon (Tokyo, Japan) TMS epifluorescence microscope.

### 2.3. Cell Morphology

After RSL3 treatment, cells were washed with modified Krebs buffer and imaged in a phase-contrast Nikon microscope. Subsequently, images were segmented using ImageJ version 1.54g (https://imagej.net/ij) creating binary masks. From the resulting segmentation images, cell area, perimeter, and circularity (a measure of roundedness) were determined.

### 2.4. IP_3_R1 Downregulation and Mitochondrial Calcium Detection

Cover-glass-grown cells were transfected using Lipofectamine 2000 (Thermo Fisher Scientific) and DNA at a 1:3 ratio. Cells were incubated for 6 hours (h) with the Lipofectamine/DNA complex. For IP_3_R1 downregulation, cells were transfected with PTRIPZ (Dharmacon, Lafayette, CO, USA) carrying a shRNA sequence against the type-1 IP_3_R (IP_3_R1) isoform and a red fluorescent (RFP) reporter sequence. Forty-eight hours after transfection with PTRIPZ, cells were challenged with RSL3 (5 µM, 4 h), and intracellular calcium was determined as described below. After calcium recording, cells were imaged to individualize transfected cells and their morphology, including their roundness degree.

For mitochondrial calcium sensing, cells were transfected with the plasmid pCMV CEPIA2mt (Addgene, https://www.addgene.org), carrying the mitochondrial calcium sensor Cepia2 [28]. Similarly, mitochondrial calcium levels in pCMV CEPIA2mt-transfected cells were determined 48 h after transfection.

### 2.5. Western Blot Analysis

After the corresponding treatments, SH-SY5Y cells were treated for 15 min with Radioimmunoprecipitation Assay (RIPA) lysis buffer supplemented with proteinase inhibitors (Waltham, MA, USA). After centrifugation, supernatants were collected, analyzed by SDS-polyacrylamide gel electrophoresis, and samples were wet transferred to nitrocellulose membranes. Membranes were blotted overnight at 4 °C using the following primary antibodies: rabbit anti-IP_3_R1 (1/1000, Cell Signaling Technology, Danvers, MA, USA), mouse anti-β-actin (1/5000, Sigma-Aldrich, St. Louis, MO, USA), rabbit anti-cleaved Caspase-3 (1/1000, Cell Signaling Technology), rabbit anti-Bcl-2 (1/1000, Santa Cruz Biotechnology, Dallas, TX, USA). For the detection of RyR channels, Western blot analysis of SH-SY5Y cells was performed as previously described [29]. Samples were separated by electrophoresis in 3.5–8% Tris-Acetate gels; gels were immersed in Tris-Tricine buffer and run for the first hour at 80 mV and then for the following 2 h at 100 mV. The protein bands were transferred to PDVF membranes (Millipore Corp. Burlington, MA, USA) and were incubated for 1 h at room temperature using as blocking solution Tris-buffered saline (TBS) with 5% fat-free milk for RyR2 and IP_3_R detection, or 5% bovine serum albumin (BSA) for RyR1 and RyR3 detection. The membrane was incubated overnight at 4 °C in blocking buffers with specific primary antibodies anti-RyR1 (1:500, generously provided by Dr. Vincenzo Sorrentino), anti-RyR2 (1:3000, MA3916, Invitrogen, Carlsbad, CA, USA), anti-RyR3 (1:2000, ab9082, Sigma Aldrich), and anti-IP_3_R (1:2500, PA1-901, ThermoFisher Scientific). Membranes were washed with saline and were then incubated for 2 h at room temperature with anti-rabbit or anti-mouse IgG antibodies coupled to radish peroxidase (1/5000, Thermo Fisher Scientific). Image acquisition was performed with the Image Lab software Version 6.0, Chemidoc TM MP System (Bio-Rad #12003154, Hercules, CA, USA).

### 2.6. ROS Detection

The production of ROS was detected by incubation of cells with 5 µM H2DCFDA (Thermo Fisher Scientific), which emits fluorescence when oxidized to dichlorofluorescein (DCF) [30]. An often-unrecognized fact is that DHDCF is a highly selective hydroxyl radical sensor [31].

### 2.7. Calcium Change Recordings

For fura-2-based cytoplasmic calcium determinations, cells were loaded for 30 min with 3 µM fura-2-AM in modified Krebs buffer at 37 °C. Afterwards, cells were washed twice with PBS and were kept in dark for 10 min before starting the recordings. The emitted fluorescence at 510 nm was recorded in a fluorimeter, alternating the excitation between 340 and 380 nm. Recordings lasted 6 min, where the first 2 min corresponded to basal fluorescence determination. Then, 0.01% Triton X-100 was added in order to permeabilize the cell membrane and saturate the probe, and the record was continued for additional 2 min. Finally, 50 mM EGTA was added to determine the fluorescence of the calcium-unbound probe. Calcium concentrations were calculated using the Grynkiewicz equation [32]:[Ca^2+^]_i_ (nM) = Kd × [(R − R_min_)/R_max_ − R)] × Sfb
where Kd = 145 nM is for fura-2 at 20 °C [33]; R is the ratio between the fluorescence produced by the excitation at 340 nm (bound) and 380 nm (unbound); R_max_ corresponds to the fluorescence ratio emitted by the calcium-saturated probe (after Triton X-100 treatment) and R_min_ is the ratio of fluorescence emitted by the calcium-depleted probe (EGTA); Sfb is the ratio of the fluorescence recorded at 380 (unbound probe) after Triton-X-100 and EGTA additions [34].

For fluo-3 based calcium imaging, cells were loaded for 30 min with 5 µM fluo-3-AM in modified Krebs buffer at 37 °C. Then, cells were washed twice with PBS and left to rest for 10 min before recordings. Recordings were performed in an LSM 710 Zeiss (Carl Zeiss AG, Oberkochen, Germany) confocal microscope using the 488 nm laser for fluo-3 excitation. Setup configuration was kept constant for every condition in each experiment.

### 2.8. Lipid Peroxidation

To assess lipid peroxidation after RSL3 treatment, cells were loaded with 5 µM BODIPY C11 (Thermo Fisher Scientific) at 37 °C for 30 min. Then, cells were washed twice with PBS and fixed for 15 min with 4% paraformaldehyde at 4 °C. Subsequently, cells were washed three times with PBS and were mounted in microscope slides to be imaged in a LSM 710 Zeiss confocal microscope.

For detection of 4-hydroxy-2-nonenal (HNE)-protein adducts, cells were fixed with 4% paraformaldehyde (PFA), followed by a permeabilization with 0.1% Triton-X-100 for 30 min and blocked with 4% BSA for 30 min. Cells were incubated with anti-HNE (1/400, Abcam, Waltham, MA, USA) at 4 °C overnight, washed three times with PBS, and incubated with an Alexa 488 anti-mouse secondary antibody (1/400, Thermo Fisher Scientific). Cells were washed two times with PBS and mounted in microscope slides to be imaged in an LSM 710 Zeiss confocal microscope.

### 2.9. Statistical Analysis

The Shapiro–Wilk test was used to determine the normal distribution of replicates. For the comparison of multiple experimental conditions, one-way ANOVA was used to test for differences in mean values, and Dunnett’s post hoc test was used for comparisons between mean values. For the comparison of two experimental conditions, the unpaired two-tail Student’s t test was used to compare differences between mean values. A *p* value < 0.05 was taken as statistically significant.

## 3. Results

### 3.1. RSL3 Induces Ferroptosis in SH-SY5Y Neuroblastoma Cells

The effects of RSL3 in inducing ferroptosis in SH-SY5Y neuroblastoma cells have not been systematically characterized to date [35], so we first determined the ferroptotic characteristics of RSL3-induced SH-SY5Y cell death.

#### 3.1.1. RSL3 Treatment Induces Changes in Cell Morphology and Time- and Concentration-Dependent Cell Death

The viability of SH-SY5Y cells was reduced as a function of both the concentration and time of incubation with RSL3 (Figure 1A). In addition, RSL3 markedly affected cell morphology (Figure 1B, upper panel). Following RSL3 treatment, a number of cells detached from the substrate, showing a birefringent aspect (Figure 1B, top panels). The assessment of cell viability by PI stain showed a large increase in PI-stained non-viable cells in RSL3-treated cultures (Figure 1B, lower panels, and Figure 1C).

Following the above observations, we evaluated three morphological criteria that turned out to be rapidly altered after treatment with RSL3, including the cell area, the perimeter, and the circularity (relationship between the dimensions of the perpendicular axes of the cell) (Figure 2).

We noted that the change in morphology was asynchronous. After 3 h of RSL3 treatment, it was possible to distinguish cells with intact morphology, others in the process of rounding off but clearly attached to the substrate, and others completely rounded (Figure 2A). Cell morphology analysis with the ImageJ program revealed that, compared to control cells, RSL3-treated cells had reduced surface area, a smaller perimeter, and a larger degree of circularity (Figure 2B).

#### 3.1.2. RSL3 Induces ROS Production and Lipid Peroxidation in SH-SY5Y Neuroblastoma Cells

To assess the ferroptotic characteristics of RSL3-treated cells, we determined markers of ROS production and membrane oxidation (Figure 3).

The generation of ROS was evaluated via DCF fluorescence. The addition of 20 µM RSL3 induced a robust ROS production, similar to that elicited by 250 µM H_2_O_2_ (Figure 3A). We then evaluated lipid peroxidation levels after 3 h of RSL3 treatment using the BODIPY C11 probe, which is a long-chain oxidizable molecule that inserts into cell membranes. In the reduced state, this probe emits red fluorescence (λ emission 590 nm); however, when oxidized, its emission/excitation pattern changes towards green fluorescence (λ emission 510 nm). We found that RSL3 treatment for 3 h produced a significant increase in the ratio of green fluorescence to red fluorescence compared to the control, indicative of lipid peroxidation (Figure 3B,C). Lipid peroxidation was further assessed by determining the formation of HNE-protein adducts; 4-HNE is a highly reactive aldehyde resulting from the lipid peroxidation of polyunsaturated fatty acids such as linoleic and arachidonic. We found that treatment with RSL3 increased the formation of protein-HNE adducts, and that co-incubation with the iron chelator deferoxamine (DFO) circumvented this increase (Figure 3D,E).

#### 3.1.3. RSL3 Induces Non-Apoptotic Cell Death in SH-SY5Y Neuroblastoma Cells

To ascertain the non-apoptotic nature of RSL3-mediated SH-SY5Y cell death, the levels of two apoptosis-related proteins, Bcl-2 and cleaved Caspase-3, were determined (Figure 4).

We found that RSL3 treatment for 6 h or 8 h significantly decreased Bcl-2 protein levels (Figure 4A,B), whereas Caspase-3 was detectable at very low levels only after 8 h of treatment with RSL3 (Figure 4A). Treatment with thapsigargin for 16 h served as a positive control for the induction of apoptosis, allowing comparison of the levels of cleaved caspase-3 in comparison to those induced by RSL3 treatment for 8 h. The levels of cleaved caspase-3 were 12–14-fold higher after treatment with thapsigargin, as compared to RSL3 treatment for 8 h (Figure 4C). Based on the cleaved caspase-3 levels, it appears that in SH-SY5Y cells, RSL3 induces mostly non-apoptotic cell death, with the caveat that at longer times of incubation (6–8 h), apoptosis markers are observed (decreased Bcl-2 protein levels).

Overall, the data substantiate the notion that SH-SY5Y cells treated with RSL3 is a validated model for the study of ferroptosis.

### 3.2. Treatment with RSL3 Causes a Delayed Increase in Intracellular Calcium Concentration

Since RSL3 induced an increase in ROS, plus the fact that the intracellular calcium channels RyR and IP_3_R are activated by oxidation [23,24,25,26], the possible effects of RSL3 on increasing the intracellular calcium concentration in SH-SY5Y cells were investigated. To measure the intracellular calcium concentration, we used fura-2, a ratiometric calcium probe that has different fluorescence absorption in its calcium-bound or -unbound states [33]. Calcium-free fura-2 has an excitation maximum at 340 nm, whereas when bound to calcium, this maximum shifts to 380 nm. Therefore, it is possible to normalize the calcium concentrations with respect to the amount of loaded probe, exciting at the two excitation wavelengths and recording at 510 nm. Preliminary studies did not detect RSL3-induced increases in intracellular calcium at short (0–20 min) time periods, which is in line with the calcium dynamics observed in RSL3-induced ferroptosis in primary hippocampal neurons [27]. To assess whether calcium concentrations were altered at longer (hours) times, cells were treated with RSL3 for different times and then loaded with fura-2 (Figure 5A).

A gradual increase in cytoplasmic calcium concentration was found in cells treated with 5 µM RSL3, from about 100 nM in the untreated situation to 300 nM after 4 h of RSL3 treatment, with significant differences at 3 h and 4 h post RSL3 addition (Figure 5A). The RSL3-induced cytoplasmic calcium increase was further explored by treating cells with RSL3 for different times and then loading them with fluo-3, a calcium sensor with higher Kd than fura-2 [33]. The fluorescence intensity of fluo-3 increased in proportion to the duration of the previous RSL3 treatment (Figure 5B,C).

To further typify the RSL3-induced intracellular calcium upsurge, cells were treated for different times with RSL3 and then challenged with the calcium ionophore ionomycin to allow for calcium influx from the extracellular medium. It was found that fluo-3 responded to ionomycin with increased fluorescence (Figure 5D,E). In addition, it was observed that the response was inversely related to the pre-incubation time with RSL3, a result consistent with an increased intracellular calcium concentration as a function of increased times of RSL3 treatment.

Due to the relevance of lipid peroxidation in ferroptosis, a possible causal relationship between RSL3-induced calcium upsurge and lipid peroxidation was investigated. To this end, we determined the effects of BAPTA-AM on the RSL3-induced oxidation of BODIPY C11. As previously carried out, we also measured the formation of 4-HNE-protein adducts after RSL3 treatment.

We found that pre-incubation with BAPTA-AM largely prevented RSL3-induced BODIPY oxidation (Figure 6A,B). Likewise, we found that the levels of 4-HNE adducts were significantly lower in BAPTA-AM-treated cells (Figure 6C,D). Thus, it is apparent that mechanistically, RSL3-induced intracellular calcium increase precedes lipid peroxidation.

### 3.3. The IP_3_R Calcium Channel Mediates RSL3-Induced Increase in Intracellular Calcium

We next investigated the possible participation of calcium channels on the RSL3-induced increase in intracellular calcium concentration (Figure 7). Veparamil and Nifedipine, two L-type voltage-gated calcium channel blockers [36,37], and CoCl_2_, a nonspecific calcium channel antagonist, did not afford protection against RSL3 treatment (Figure 7A). In another series of experiments, the extracellular calcium chelator EGTA and the cell-permeant calcium chelator BAPTA-AM were tested as possible protectors of RSL3 toxicity. In these series, the addition of EGTA to the culture medium failed to diminish the effects on viability due to RSL3 treatment (Figure 7B). In contrast, loading cells with BAPTA-AM prior to RSL3 treatment significantly protected SH-SY5Y cell viability (Figure 7B). As a positive control, protection by DFO was also observed (Figure 7B).

Since viability was not protected by compounds that hinder calcium influx from the extracellular medium, it is likely that the observed calcium increases induced by RSL3 treatment depend on calcium release from intracellular calcium stores. Considering that (i) the endoplasmic reticulum (ER) is the most important store of intracellular calcium, (ii) the release of calcium from this organelle is mainly regulated by two channels, the RyR and the IP_3_R channels, and (iii) that the intracellular calcium channels RyR and IP_3_R are activated by oxidation [23,24,25,26], the putative mediation by these channels in cell death induced by RSL3 in SH-SY5Y cells was evaluated (Figure 8).

No effects on RSL3-mediated cell death were observed when cells were pre-incubated for 1 h with 100 µM ryanodine, a specific RyR channel inhibitor at µM concentrations (Figure 8A). This result agrees with the fact that the SH-SY5Y cell line used in this work does not express RyR channels (Appendix A). In contrast, treatment with xestospongin B, a powerful and specific IP_3_R channel blocker, protected cells from RSL3-induced cell death (Figure 8A). The participation of the IP_3_R channel in this process was further evaluated with carbachol, a cholinergic agonist that reduces IP_3_R protein levels in SH-SY5Y cells [38]. Pre-incubation with Cch effectively produced a time-dependent decrease in IP_3_R1 levels (Figure 8B,C) and protected cells from RSL3-induced cell death (Figure 8A). Overall, these results indicate that RSL3 induces an increase in intracellular calcium that is mediated, in large part, by IP_3_R channels, and that this increase is germane to RSL3-induced ferroptotic cell death.

To further confirm the participation of IP_3_R channels in the calcium increase observed in response to RSL3 treatment, the expression of IP_3_R1, the main isoform expressed in SH-SY5Y cells [39], was downregulated. To this end, cells were transiently transfected with a plasmid containing a shRNA sequence against IP_3_R1 tagged to a RFP sequence. Transfected cells were treated, or not, with RSL3 and then loaded with fluo-3 to analyze cytoplasmic calcium levels. Fluo-3 fluorescence intensity was determined in RFP-positive (RFP+) cells, which expressed the interfering sequence and RFP-negative cells (RFP-), that did not express the shRNA sequence.

Under control conditions, no differences between RFP+ and RFP- cells were found, with a consistently low fluo-3 signal. When both groups were compared after treatment with RSL3, the fluo-3 fluorescence intensity was significantly lower in cells with knockdown for IP_3_R1 (Figure 9B), an indication that IP_3_R1 mediates the RSL3-induced calcium upsurge.

Because of a low efficiency (5–7%) of transfection with the shRNA sequence against IP_3_R1, it was not possible to evaluate RSL3-induced ferroptotic death in IP_3_R1 shRNA-transfected cells using the MTT assay, which was used to generate the results described in Figure 1. As an alternative, RSL3-induced roundness, a bona fide ferroptosis indicator, was evaluated (Figure 9C). The data indicated that IP_3_R1 shRNA-transfected cells were significantly more resistant to the change towards roundness induced by RSL3 than non-transfected cells. IP_3_R1 shRNA-transfected cells were also resistant to the RSL3-induced changes in area and perimeter described in Figure 2 (Appendix A). Overall, the data strongly suggest that IP_3_R1 mediates the ferroptotic events induced by RSL3.

### 3.4. RSL Treatment Induces an Increase in Mitochondrial Calcium

Since through mitochondria-associated ER membranes (MAMs), IP_3_R participates in calcium entry into mitochondria [40], the relative levels of mitochondrial calcium after treatment with RSL3 were investigated using the genetically encoded mitochondria calcium probe CEPIA2mt (Figure 10).

Cells treated with RSL3 showed a striking 3.5-fold increase in mitochondrial calcium levels (Figure 10A,B). Overall, these results indicate that RSL3-induced ferroptosis in SH-SY5Y cells provokes an increase in cytoplasmic and mitochondrial calcium levels that are mediated by the IP_3_R channel.

## 4. Discussion

The participation of calcium in ferroptosis is a matter of debate. Initial studies using HT-1080 epithelial cells discarded a role for calcium under the observation that ferroptosis was not blocked by the extracellular calcium chelator EGTA or by the intracellular chelator BAPTA-AM [2]. Nonetheless, other reports proposed that ferroptosis and oxytosis are the same phenomenon [20,21]. Because oxytosis is highly dependent on calcium entry into the cell, here, we evaluated the participation of calcium in a model of RSL3-induced ferroptosis using the dopaminergic cell line SH-SY5Y. Of note, this is the first report that establishes a firm link between calcium and ferroptosis in this neuroblastoma cell line. In addition to previous works, which reported that plasma membrane calcium channels [21,41] or RyR channels [27,41] act as mediators in the increase in intracellular calcium induced by RSL3, here, we additionally identified IP_3_R channels as mediators of the calcium increase induced by RSL3.

We found that RSL3 affected the viability of SH-SY5Y cells in a dose- and time-dependent manner. Thus, RSL3 induced 50% cell death at times as short as 4 h. Treatment with RSL3 also induced landmarks of ferroptosis, such as an increase in ROS levels and a remarkable change in cell morphology, with a significant increase in cell roundness. Arguably, the changes in ROS production and cell morphology are a preamble to cell death. Albeit the direct inhibitory effects of RSL3 as an inhibitor of GPx4 have been recently questioned based on studies performed in cell-free systems [42], our results support the role of RSL3 as an effective ferroptosis-inducing agent.

Associated with ROS production, there was an increase in membrane lipid peroxidation, a hallmark of ferroptosis. In accordance with the participation of iron in ferroptosis, co-incubation with the iron chelator DFO precluded lipid peroxidation, as detected by the levels of 4-HNE-protein adducts. Overall, these results indicate that RSL3 induces a robust ferroptotic response in SH-SY5Y cells.

In consideration of a putative participation of calcium in ferroptosis, we used fluorescent probes to detect possible alterations in calcium levels produced by RSL3. Firstly, we observed that the application of RSL3 did not trigger an immediate (seconds–minutes) increase in intracellular calcium concentration in the way that, for example, thapsigargin, carbachol, or even hydrogen peroxide do in N2a cells and primary hippocampal neurons [43]. However, examining at later times (longer than 1 h), we identified a gradual increase in intracellular calcium. A similar observation was reported for hippocampal cells in primary culture [27]. Using fura-2, we calculated that the basal calcium concentrations of around 100 nM, a value relatively consistent to that reported in the literature for these cells [44] increased to 300 nM after RSL3 treatments. These calcium concentrations are not particularly high when compared to the increases generated by the calcium ionophore ionomycin (≥500 nM). However, their maintenance over time could have consequences for cellular homeostasis.

An observation of interest was that the intracellular calcium chelator BAPTA-AM had a protective effect against RSL3 treatment. This protection was not observed in RSL3-treated cells incubated in a medium containing the extracellular calcium chelator EGTA or plasma membrane-resident calcium channels inhibitors. We conclude that, different from oxytosis, in the SH-SY5Y cell ferroptosis model, the increase in calcium levels induced by RSL3 has an intracellular origin.

Considering that the most prominent intracellular calcium reservoir corresponds to the ER [45], we evaluated the possible protective effect of inhibitors of the main channels that allow for calcium efflux in the ER, the RyR, and the IP_3_R calcium channels. We found that xestospongin B, an inhibitor of IP_3_R channels, but not ryanodine, an inhibitor of RyR channels, protected against RSL3-induced cell death in viability assays. The lack of effect of ryanodine agrees with the absence of RyR channels reported here for these cells, and with previous reports that found no or very little RyR protein expression in SH-SY5Y cells, albeit RyR mRNA expression was detected [46,47,48]. Notwithstanding, inhibitory concentrations of ryanodine partially protected primary hippocampal neurons against RSL3-induced death [27]. Thus, calcium release from the ER participates in two different neuronal models of ferroptotic death.

We found that preincubation with carbachol, which produces a decrease in IP_3_R protein levels [38], also exerts protection against the calcium concentration increase produced by RSL3, suggesting that IP_3_R channels mediate this increase. In the same vein, cells transfected with an interfering construct for IP_3_R1, the isoform mostly expressed in this cell line [49], presented lower calcium increases after RSL3 treatment, which is consistent with the participation of IP_3_R1 in this calcium increase. In addition, transfected cells did not undergo the morphology changes induced by RSL3, an indication that these cells are resistant to RSL3-induced ferroptosis.

Since IP_3_R is a highly regulated channel, its activation by RSL3 treatment could have several possible mediators. For example, the interaction between Bcl-2 and IP_3_R is well documented, with the former acting as an inhibitor of the latter, through interaction via the BH-4 domain of Bcl-2 [50]. We observed that RSL3 induced a decrease in Bcl-2; therefore, it is possible that a decreased inhibition of IP_3_R by Bcl-2 may be one of the causes of the higher levels of cytoplasmic calcium. In addition, the increased ROS levels produced by RSL3 are likely to enhance IP_3_R-mediated calcium release, since IP_3_R channels increase their activity in response to oxidation [23,51].

A crucial finding of this work was that RSL3 treatment induced calcium loading in mitochondria. This increase in mitochondrial calcium is a novel observation that is worthwhile to assess in other ferroptosis models. The elevation of mitochondrial calcium levels can stimulate the Krebs cycle and the electron transport chain, increasing the leakage of electrons and ROS generation [52]. In fact, it was recently demonstrated that after exposure to erastin, hyperpolarization of the inner mitochondrial membrane and greater production of mitochondrial ROS occur, which further contribute to enhancing lipid peroxidation [53]. A putative counterpart is found in a recent report that showed that in a cold stress model of ferroptosis, activation of the mitochondrial calcium uniporter MCU by the mitochondrial calcium uptake regulator MICU1 is required for lipid peroxidation and subsequent ferroptosis [54].

We observed that BAPTA-AM inhibited the lipid peroxidation induced by RSL3. Therefore, it is possible that calcium also participates in activating agents that contribute to the generation of lipid peroxides as, for example, the LOX enzymes, a group of enzymes widely linked to the initiation and development of ferroptotic death [55,56]. The membrane localization and catalytic activity of ALOX-5 and ALOX-15 are highly calcium dependent [57,58,59]. Arguably, RSL3-induced calcium increases could stimulate lipid peroxidation through this mechanism. This proposal is in line with previous findings in NIH-3T3 cells, which reported that increases in cytoplasmic calcium concentrations occurred prior to plasma membrane rupture [60].

A notorious difference between oxytosis and ferroptosis is that in oxytosis, calcium enters the cytoplasm from the extracellular space, in a process known as store-operated calcium entry (SOCE) [21,61], whereas in the ferroptosis model described here and in a primary hippocampal cell model previously described by us [27], calcium is released into the cytoplasm from an internal store, i.e., the ER.

Overall, the data indicate that inhibition of GPx4 by RSL3 generates a vicious circle between calcium and ROS increase, as depicted in Figure 11. An initial increase in ROS is generated either directly by the increase in lipid peroxides, through the axis LOX → lipid peroxide → Bid → mitochondrial dysfunction [15], or through some other undefined mechanism. The increased oxidation tone activates the IP_3_R with the ensuing increase in cytosolic calcium and increased mitochondrial calcium uptake by MCU, which, in turn, would intensify mitochondrial dysfunction by stimulating the production of ROS. Cell death occurs through the accumulation of lipid peroxidation damage and the decreased synthesis of ATP, which is needed for repair processes. An alternative source of IP_3_R stimulation, shown at the bottom of Figure 11, is through the activation of phospholipase C by lipid peroxides [62], with the subsequent activation of IP_3_R by IP_3_.

## 5. Conclusions

In summary, the results presented here point to an IP_3_R-mediated calcium increase as a mediator of RSL3-induced ferroptosis in SH-SY5Y cells. The generation of a vicious cycle between calcium and mitochondrial ROS production will contribute to the generation of toxic levels of lipid peroxides that, finally, will result in ferroptotic cell death.

## Figures and Tables

**Figure 1 antioxidants-13-00196-f001:**
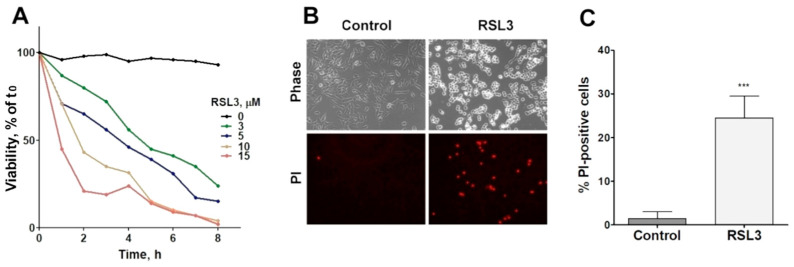
RSL3 affects the viability of SH-SY5Y cells. (**A**) Cells were treated with different concentrations of RSL3 for different times and viability was evaluated with the MTT assay. Viability at time = 0 was normalized to 100%. Values represent means of 5 replicates per experimental point of a representative experiment. N = 3 independent experiments. (**B**) Cells were treated for 3 h with 5 µM RSL3 or vehicle (DMSO, Control) and were stained with propidium iodine (PI, red) to evaluate dead cells in the population. Representative images are shown. Scale bar: 80 µm. (**C**) Quantification of PI-positive cells. Values represent the mean ± SD from 200–250 cells per experimental condition; *** *p* < 0.001.

**Figure 2 antioxidants-13-00196-f002:**
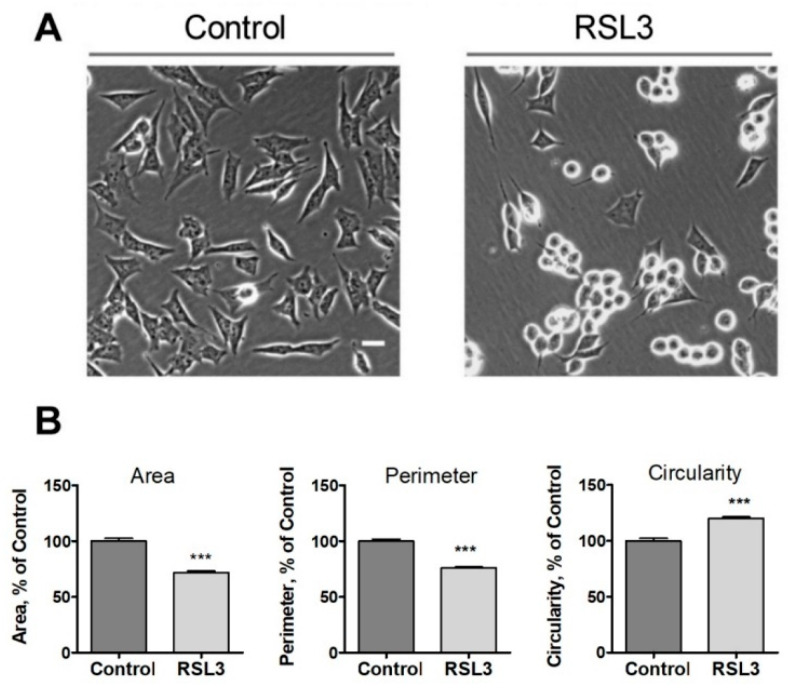
Effects of RSL3 on cell morphology. Cells were treated for 3 h with 5 µM RSL3 or vehicle and then photographed using phase contrast. (**A**) Representative images of Control and RSL3 treatment conditions. Scale bar 20 µm. (**B**) Evaluation of area, perimeter, and roundness parameters. Values represent mean ± SEM. Between 62 and 123 cells were evaluated for each experimental condition; N = 3 independent experiments; *** *p* < 0.001.

**Figure 3 antioxidants-13-00196-f003:**
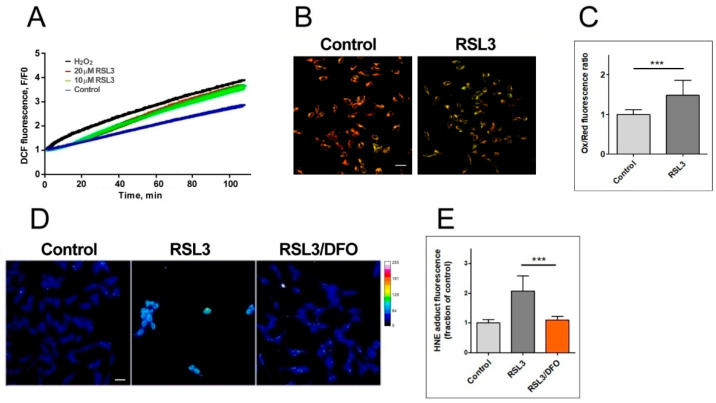
RSL3 induces an increase in ROS production and lipid peroxidation. (**A**) SH-SY5Y cells were loaded with H2DCFDA and were then treated at t = 0 with 10 µM or 20 µM RSL3 or 250 µM H_2_O_2_; DCF fluorescence intensity was recorded over time. Values represent means of 5 replicates per experimental condition. N = 3 independent experiments. (**B**) SH-SY5Y cells, treated for 3 h with 5 µM RSL3 or vehicle, were stained with BODIPY C11. Representative images of the overlapping values collected by the red (reduced, 590 nm emission) and green (oxidized, 510 nm emission) fluorescence channels for each condition are shown. Scale bar 20 µm. (**C**) Quantification of the ratio of the green to red fluorescence intensity. Values represent mean ± SD of 150 cells; *** *p* < 0.001. (**D**) Cells were treated for 4 h without (control) or with 5 µM RSL3. Representative frames of HNE adducts immunofluorescence are shown. Scale bar 20 µm. The upper right corner shows a thermal fluorescence intensity scale. (**E**) Quantification of HNE fluorescence intensity. Values represent mean ± SD for 100–120 cells per experimental condition, N = 3 independent experiments, *** *p* < 0.001.

**Figure 4 antioxidants-13-00196-f004:**
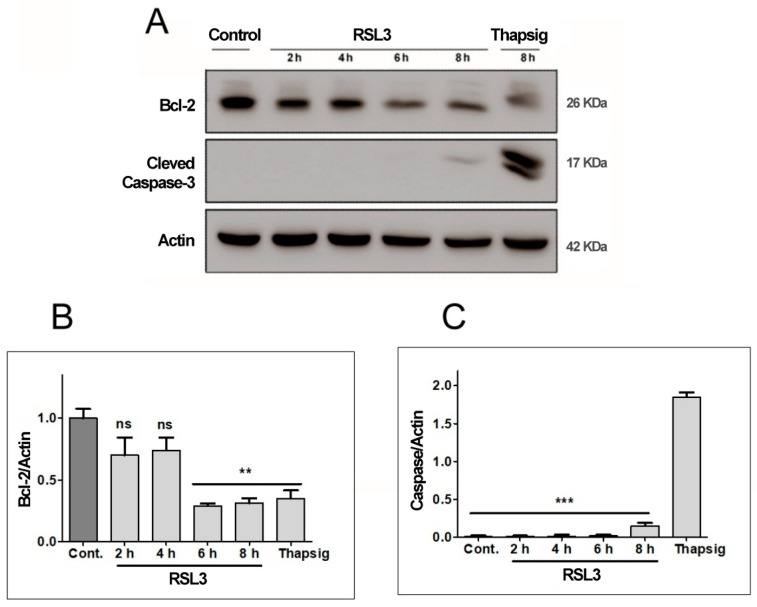
RSL3 decreases Bcl-2 levels but does not induce Caspase-3 cleavage. Extracts of cells treated with 5 µM RSL3 for different times, or with 5 µM thapsigargin (Thapsig) for 8 h, were analyzed by Western blot to evaluate Bcl-2 and cleaved Caspase-3 levels. Actin was used as a loading control. (**A**) Image of a representative blot. (**B**) Densitometric quantification of Bcl-2/Actin levels normalized to control. (**C**) Densitometric quantification of cleaved Caspase-3/Actin levels normalized to control. after treatment with RSL3 for different times, or with 5 µM thapsigargin for 16 h. Mean ± SEM values are shown. N = 3 independent experiments; ns = not significant, ** *p* < 0.01, *** *p* < 0.001.

**Figure 5 antioxidants-13-00196-f005:**
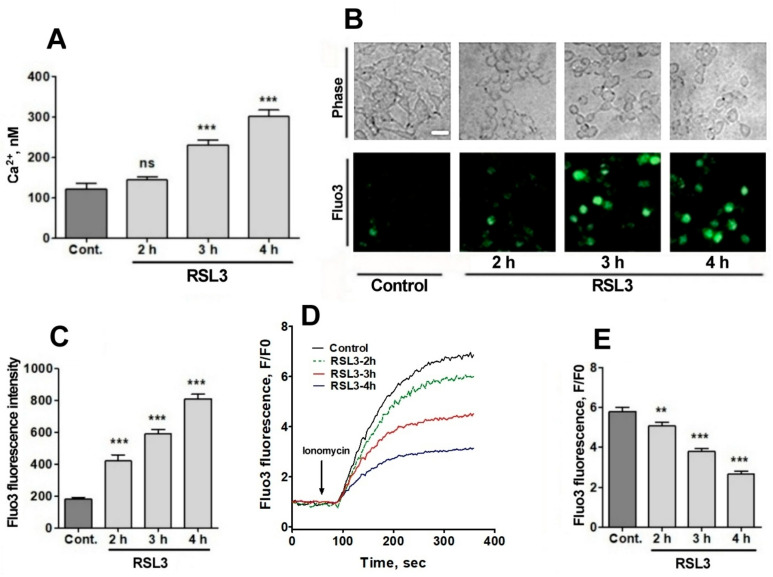
RSL3 alters calcium concentrations after a delay period. (**A**) Cells pre-treated with 5 µM RSL3 for different times were loaded with fura-2 and fluorescence was recorded to subsequently determine cytoplasmic calcium concentrations. Values represent mean ± SEM of a representative experiment from 3 independent experiments with 3 replicates per experimental condition. Differences were determined by one-way ANOVA followed by Dunnett’s post hoc test. (**B**) Cells pre-treated with 5 µM RSL3 or vehicle (DMSO, control) were loaded with fluo-3 and fluorescence was recorded. Representative images of each treatment are shown. Scale bar 20 µm. (**C**) Quantification of cell fluorescence after the different treatments. Values are given as mean ± SEM; 40 cells per condition. (**D**) Plots of the ratio of fluorescence intensity over initial fluorescence (F/F0) as a function of time, before and after the addition of ionomycin (arrow). (**E**) Maximum F/F0 ratio reached after the addition of ionomycin for each condition. Mean ± SEM of the last three points for each experimental condition with points in triplicate. ns = not significant; ** *p* < 0.01; *** *p* < 0.001.

**Figure 6 antioxidants-13-00196-f006:**
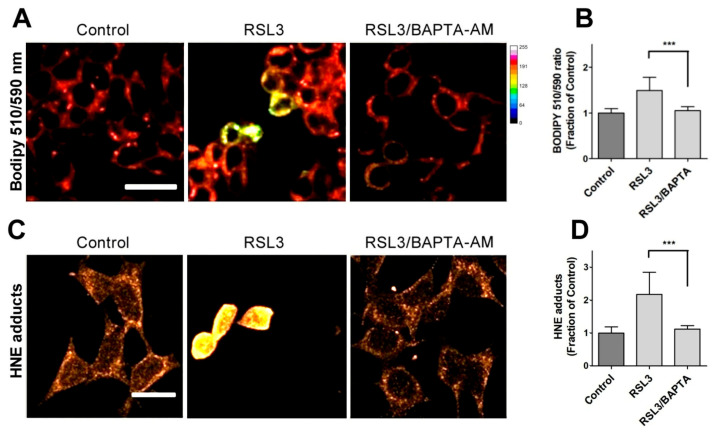
Calcium chelation protects cells against RSL3-induced lipid peroxidation. Cells were pre-incubated for 1 h with 5 µM BAPTA-AM followed by co-incubation for 3 h with 5 µM RSL3 and analysis of lipid peroxidation with either BODIPY C11 or anti-HNE immunofluorescence as described in Methods. (**A**) Representative images of BODIPY fluorescence upon treatment with RSL3, without (control) or after pre-treatment with BAPTA-AM. Scale bar 20 µm. (**B**) Quantification of the ratio of fluorescence intensity of the oxidized (green) channel to that of the reduced (red) channel. At least 80 cells per condition were quantified from N = 3 independent experiments. Mean ± SD values are shown. (**C**) Representative images of immunodetection against 4-HNE adducts. Scale bar 15 µm. (**D**) Quantification of fluorescence intensity. Values represent mean ± SD from 75–90 cells quantified for each experimental condition from N = 3 independent experiments. Both in (**B**,**D**) differences were evaluated by one-way ANOVA, followed by Dunnett’s post hoc test. *** *p* < 0.001.

**Figure 7 antioxidants-13-00196-f007:**
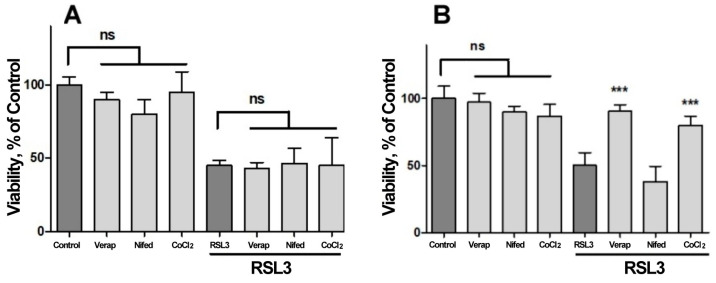
Both BAPTA and DFO protect against RSL3-induced decrease in cell viability. Cells were incubated for 90 min with 300 µM Verapamil (Verap), 5 µM Nifedipine (Nifed) or 100 µM CoCl_2_ (**A**) or with 25 µM DFO, 2.5 mM EGTA or 2.5 µM BAPTA-AM (**B**), followed by incubation for 4 h with 5 µM RSL3. After that, cell viability was estimated by the MTT assay. Values represent mean ± SEM. N = 3 independent experiments with determinations done in sextuplicate. Differences were assessed by one-way ANOVA followed by Dunnett’s test. *** *p* < 0.001 compared to RSL3 alone treatment; ns = not significant.

**Figure 8 antioxidants-13-00196-f008:**
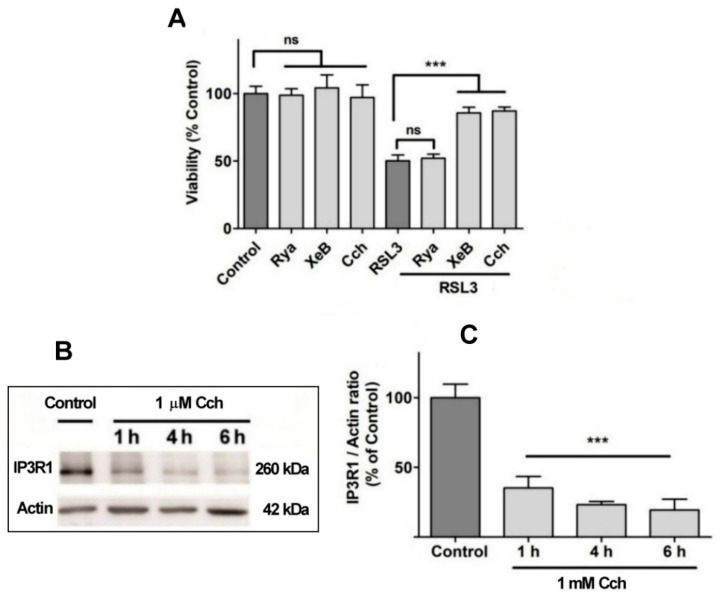
The role of IP_3_R channels in RSL3-induced ferroptosis. (**A**) SH-SY5Y cells were pre-incubated for 1 h with 100 µM Ryanodine, 10 µM xestospongin B (XeB) or 1 mM carbachol (Cch) and were then incubated for 4 h with RSL3. Viability was determined by the MTT assay; ns = not significant, *** *p* < 0.001. (**B**) Representative Western blots against IP_3_R1 after Cch treatments for 0, 1, 4 or 6 h. (**C**) Quantification of relative protein levels of IP_3_R1 after Cch treatment. Values represent the mean ± SD from N = 3 independent experiments; *** *p* < 0.001 compared to Control.

**Figure 9 antioxidants-13-00196-f009:**
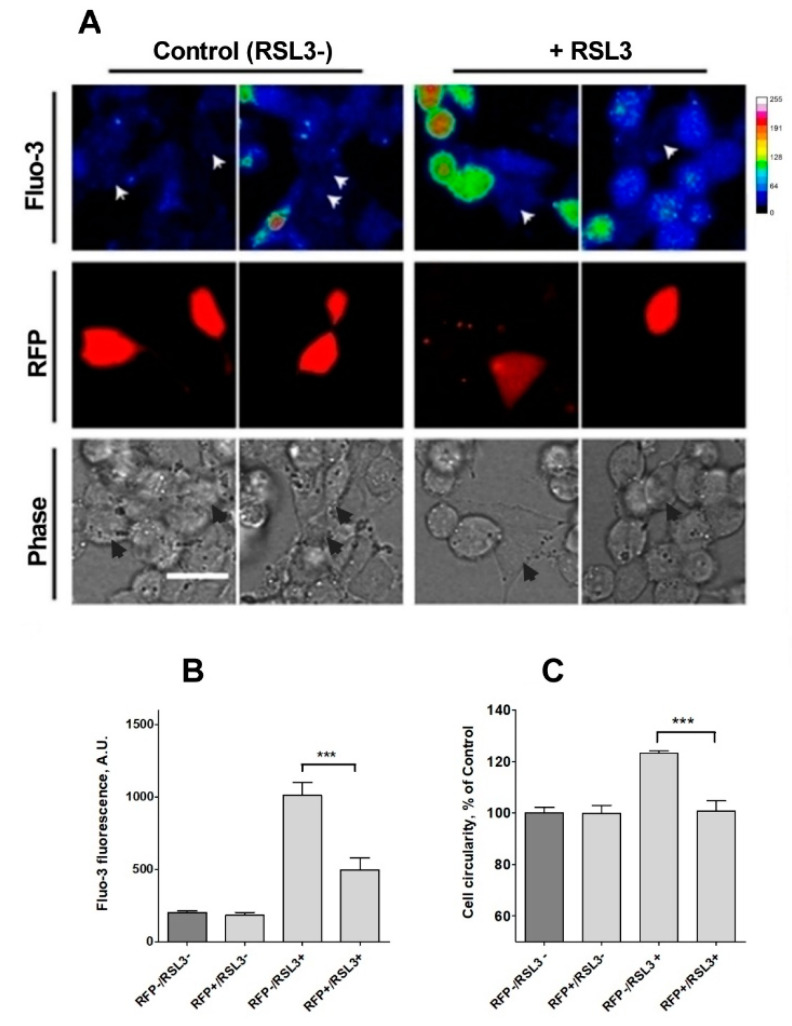
Knockdown of IP_3_R1 reduces the increase in intracellular calcium concentration induced by RSL3. Cells previously transfected with a shRNA construct against IP_3_R1 with a RFP tag (RFP+) were treated for 4 h with 5 µM RSL3 or vehicle, then loaded with fluo-3 and frames were obtained with red and green fluorescence filters. (**A**) Representative images in thermal scale of non-treated cells (columns 1 and 2) and RSL3-treated cells (columns 3 and 4). Scale bar 20 µm. White (Fluo-3) and black (Phase) arrowheads mark the position of RFP-positive cells. (**B**) Quantification of fluo-3 fluorescence in RFP+ cells, which corresponds to IP_3_R1 shRNA transfected cells, and RFP- cells, which do not express the IP_3_R1 construct, without (RSL3-) or with (RSL3+) RSL3 treatment. (**C**) Estimation of roundness in RFP+ and RFP- cells treated or not with RSL3. Values represent mean ± SEM. Between 32 and 115 cells were evaluated per experimental condition from 2 independent experiments. *** *p* = 0.001 comparing the RFP-/RSL3+ and the RFP-/RSL3+ conditions. No significant changes were detected between the RFP-/RSL3-, RFP+/RSL3- and RFP+/RSL3+ conditions.

**Figure 10 antioxidants-13-00196-f010:**
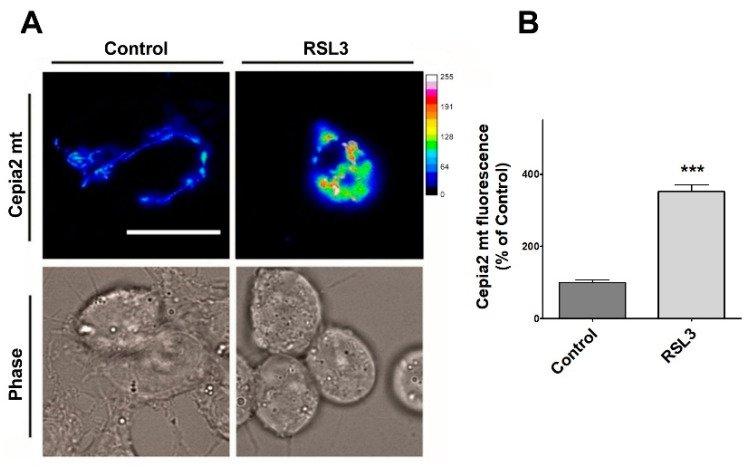
RSL3 induces an increase in mitochondrial calcium levels. Cells transfected with the mitochondrial calcium sensor CEPIA2mt were treated with 5 µM RSL3 for 4 h or with control solution, and the fluorescence intensity was recorded. (**A**) Representative images of both conditions shown in thermal scale (right-hand bar). Size bar 30 µm. (**B**) Quantification of fluorescence intensity. Values represent Mean ± SEM from 58–62 cells quantified for each experimental condition. Difference between mean values was determined by unpaired two-tail *t*-test. N = 2 independent experiments. *** *p* < 0.001.

**Figure 11 antioxidants-13-00196-f011:**
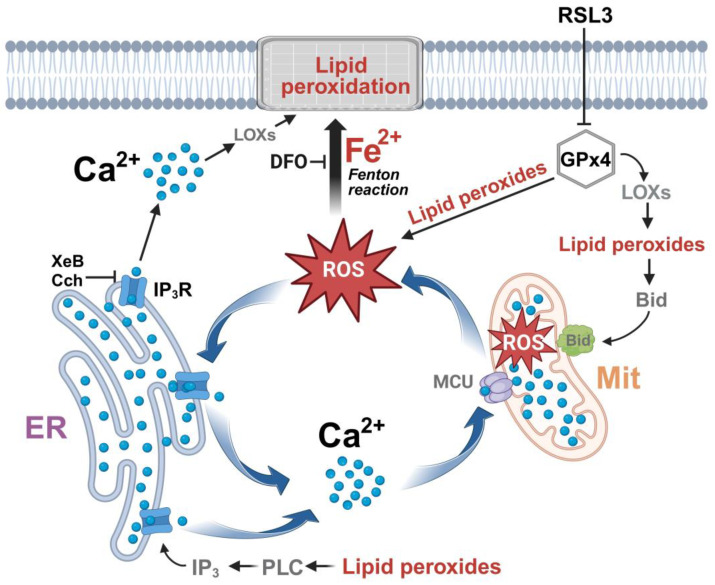
The effect of RSL3-induced ferroptosis on intracellular calcium dynamics in SH-SY5Y cells. ER, endoplasmic reticulum; Mit, mitochondria; ROS, reactive oxygen species; DFO, iron chelator deferoxamine; MCU, mitochondrial calcium uniporter; LOXs, lipoxygenases; IP_3_, inositol 1,4,5-trisphosphate; IP_3_R, IP_3_ receptor; PLC, Phospholipase C; XeB, xestospongin B; Cch, carbachol. Figure created with BioRender.com.

## Data Availability

The data presented in this study are available on request from the corresponding author.

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
