# Peer review of "IP_3_R-Mediated Calcium Release Promotes Ferroptotic Death in SH-SY5Y Neuroblastoma Cells"

_antioxidants, 2024, doi:10.3390/antiox13020196_

Round 1

Reviewer 1 Report (Previous Reviewer 1)

Comments and Suggestions for Authors

I have reviewed the contents of the supplementary file. As far as I am concerned, the authors have addressed my concerns and I have no further points. I therefore recommend the manuscript for publication in Antioxidants.

Reviewer 2 Report (Previous Reviewer 2)

Comments and Suggestions for Authors

The manuscript has been revised and been improved. No further comments.

This manuscript is a resubmission of an earlier submission. The following is a list of the peer review reports and author responses from that submission.

Round 1

Reviewer 1 Report

Comments and Suggestions for Authors

In this paper, the authors claim that IP3R-mediated calcium release promotes ferroptosis in SH-SY5Y neuroblastoma cells. However, I believe that the experimental results do not fully support this conclusion. There are two major concerns with this paper. The specific points are as follows.

Major points.

1.      Although the authors claim that IP3R-mediated calcium release promotes ferroptosis in SH-SY5Y neuroblastoma cells using inhibitors such as ryanodine (a RyR channel inhibitor) and IP3R channel blockers, this is insufficient due to specificity issues. First, it should be tested whether knockdown of IP3R in this cell line would inhibit ferroptosis. This is a minimum requirement.

2.      The fact that RSL3, a GPX4 inhibitor, induces ferroptosis (Fig. 1), changes cell morphology (Fig. 2), causes lipid peroxidation (Fig. 3) and does not activate caspases (Fig. 4) are all features of ferotosis and have no implications because they are common phenomena. The fact that calcium is also relevant has been reported in the past (Figs. 5-7). The first half of the paper is almost a meander and has nothing new to offer. The paper needs a major revision.

Minor points.

English should be carefully revised throughout the manuscript.

Author Response

Reviewer 1.

In this paper, the authors claim that IP3R-mediated calcium release promotes ferroptosis in SH-SY5Y neuroblastoma cells. However, I believe that the experimental results do not fully support this conclusion. There are two major concerns with this paper. The specific points are as follows.

Major points.

  1. Although the authors claim that IP3R-mediated calcium release promotes ferroptosis in SH-SY5Y neuroblastoma cells using inhibitors such as ryanodine (a RyR channel inhibitor) and IP3R channel blockers, this is insufficient due to specificity issues. First, it should be tested whether knockdown of IP3R in this cell line would inhibit ferroptosis. This is a minimum requirement.

Replay. We thank the reviewer for his/her observation. Because of the low degree of transfection, 5-7%, we were unable to measure directly RSL3-induced death in IP3R1 knockdown cells, e.g. by MTT assay as done in Fig. 1. Acknowledging the relevance of the reviewer´s requirement, we now introduced in Figure 9C and Supplemental Figure 2 evidence on a bona fide ferroptosis indicator: RSL3-induced changes in cell shape. The new data clearly indicates that IP3R shRNA-transfected cells are resistant to changes in roundness, area and perimeter induced by RSL3.

  1. The fact that RSL3, a GPX4 inhibitor, induces ferroptosis (Fig. 1), changes cell morphology (Fig. 2), causes lipid peroxidation (Fig. 3) and does not activate caspases (Fig. 4) are all features of ferotosis and have no implications because they are common phenomena. The fact that calcium is also relevant has been reported in the past (Figs. 5-7). The first half of the paper is almost a meander and has nothing new to offer. The paper needs a major revision.

Replay. We respectfully disagree with the opinion of the reviewer that data of Figure 1 to 4 are meander and has nothing new to offer.

As stated at the beginning of the Results section, the effects of RSL3-induced ferroptosis in SH-SY5Y neuroblastoma cells have not been systematically characterized up to date. Because of that, it was necessary to determine if RSL3 does induce ferroptotic cell death in SH-SY5Y cells. To underline the results depicted in Figure 1 to 4, we now state in lines 322-323 of the revised manuscript that “Overall, the data substantiate the notion that SH-SY5Y cells treated with RSL3 is a validated model for the study of ferroptosis”. This valuable new information provides the scientific community with a new cell model for the study of ferroptosis.

Calcium. As stated in the Introduction, the role of calcium in ferroptosis is a matter of controversy. Although the participation of calcium was dismissed in the initial studies of ferroptosis, the group of Maher and collaborators has proposed the hypothesis that ferroptosis and oxytosis are parts of the same phenomenon. Being oxytosis death calcium-dependent, it follows the ferroptosis should also be calcium-dependent. Our results not only support this hypothesis but also identifies the endoplasmic reticulum calcium channel IP3R as the mediator of ferroptotic cell death in SH-SY5Y cells. This is an absolutely novel observation that qualitatively adds to our knowledge of ferroptosis as a calcium-dependent cell death process.

Minor points.

English should be carefully revised throughout the manuscript.

Replay. We thank the reviewer for this observation. English grammar and syntax was thoroughly checked.

Reviewer 2 Report

Comments and Suggestions for Authors

The manuscript is comprehensive and is fully described. However, some typing mistakes are in the text, and certain sentences require clarification.

For instance, L48-51, L77-81, L169-170, L331-332

Questions for the authors:

1.      In the abstract L24-25, “We show that inhibition of GPx4 by RAS Selective Lethal Compound 3 (RSL3) generated an increase in reactive oxygen species(ROS),” but there is no evidence to show the inhibition of GPx4 in Results.

2. L313-316, it is unlikely 12-14 fold difference in Fig.4C. The authors stated that RSL3 induces mostly non-apoptotic cell death. Have the authors detected the proportion of total cell death, ferroptosis and non-apoptotic cell death?

Comments on the Quality of English Language

some typing mistakes are in the text, and certain sentences require clarification. For instance, L48-51, L77-81, L169-170, L331-332

Author Response

Reviewer 2

The manuscript is comprehensive and is fully described. However, some typing mistakes are in the text, and certain sentences require clarification.

Questions for the authors:

  1. In the abstract L24-25, “We show that inhibition of GPx4 by RAS Selective Lethal Compound 3 (RSL3) generated an increase in reactive oxygen species(ROS),” but there is no evidence to show the inhibition of GPx4 in Results.

Replay. We thank the reviewer for this observation. The sentence was changed to “We show that treatment with RAS Selective Lethal Compound 3 (RSL3), a GPX4 inhibitor, produced an increase in reactive oxygen species (ROS)…”

  1. L313-316, it is unlikely 12-14 fold difference in Fig.4C. The authors stated that RSL3 induces mostly non-apoptotic cell death. Have the authors detected the proportion of total cell death, ferroptosis and non-apoptotic cell death?

Replay. The levels of cleaved caspase-3 were indeed 12.3-fold higher after treatment with thapsigargin as compared to RSL3 treatment for 8 h. Data in Fig. 4C show values of 0.15 ± 0.04 and 1.85 ± 0.07 (Mean ± SEM) for RSL3 treatment for 8 h and thapsigargin treatment for 8 h, respectively.

We did not estimate the ratio between ferroptosis and non-apoptotic cell death. The ratio between ferroptosis and apoptotic cell death will depend on the time of treatment. At 8 h, this ratio was 12.3-fold.

Comments on the Quality of English Language. Some typing mistakes are in the text, and certain sentences require clarification. For instance, L48-51, L77-81, L169-170, L331-332

L48-51. The sentence was changed to “Within these cell death processes ferroptosis stands out, with over 4000 publications during 2023 alone.”.

L77-81. The sentence was changed to “Of note is the relationship between the inhibition of GPx4 and increased ROS production. A mediator between these two landmarks of ferroptosis is the pro-apoptotic protein Bid [15]. Based on the observation that the Bid inhibition preserves cell viability in RSL3-treated cells, the authors propose a sequence of events in which GPx4 inhibition enhances 12/15 LOX activity, which results in lipid peroxide formation and BID transactivation.”

L169-170. The sentence was changed to “The production of ROS was detected by incubation of cells with 5 μM H2DCFDA (ThermoFisher Sci.), which emits fluorescence when oxidized to dichlorofluorescein (DCF)”.

L331-332. The sentence was changed to “Unbound fura-2 has an absorbance maximum at 340 nm, whereas when bound to calcium this maximum shift to 380 nm.”